# Effect of Potassium Phosphate Content in Aluminosilicate Matrix on Mechanical Properties of Carbon Prepreg Composites

**DOI:** 10.3390/ma15010061

**Published:** 2021-12-22

**Authors:** Eliška Kohoutová, Pavlína Hájková, Jan Kohout, Aleš Soukup

**Affiliations:** 1ORLEN UniCRE, a.s., Revoluční 1521/84, 40001 Ústí nad Labem, Czech Republic; Pavlina.Hajkova@unicre.cz (P.H.); Jan.Kohout@unicre.cz (J.K.); Ales.Soukup@unicre.cz (A.S.); 2Department of Material Science, Faculty of Mechanical Engineering, Technical University of Liberec, Studentská 1402/2, 461 17 Liberec, Czech Republic

**Keywords:** aluminosilicate, composite, inorganic matrix, phosphate, prepreg, tensile strength

## Abstract

Six matrices based on alkali-activated aluminosilicate with different amounts of potassium phosphate were prepared for the production of six-layer composite plates. The addition of potassium phosphate in the matrix was 2 wt%, 4 wt%, 6 wt%, 8 wt% and 10 wt% of its total weight. The matrix without the potassium phosphate was also prepared. The aim of this study was to determine whether this addition has an effect on the tensile strength or Young’s modulus of composites at temperatures up to 800 °C. Changes in the thickness and weight of the samples after this temperature were also monitored. Carbon plain weave fabric was chosen for the preparation of the composites. The results show that under normal conditions, the addition of potassium phosphate has no significant effect on the mechanical properties; the highest measured tensile strengths were around 350 MPa. However, at temperatures of 600 °C and 800 °C the addition of potassium phosphate had a positive effect, with the tensile strength of the composites being up to 300% higher than the composites without the addition. The highest measured values of composites after one hour at 600 °C were higher than 100 MPa and after 1 h at 800 °C higher than 85 MPa.

## 1. Introduction

Alkali-activated aluminosilicates are one of the most researched materials in recent decades. They gain their attention with their relatively low costs, simple preparation without the need to cure at high temperatures, extensive raw material base, a wide range of uses and especially with their resulting physical and chemical properties [1,2,3].

As the name implies, the basic components of these materials are aluminosilicates and an alkaline activator. Aluminosilicates are most often represented by metakaolin, but the content of aluminum and silicon can also be supplied in the form of industrial waste, such as fly ash or blast furnace slag. These alternative materials give rise to a more environment friendly and cost-effective inorganic material [4,5,6,7]. The liquid alkaline activator in which the aluminosilicate powder is dissolved is most often represented by a solution of water glass, oxides and hydroxides of sodium or potassium or their mixtures [4,8].

The properties of the inorganic binders described above depend on the type and amount of each component. In order to improve some properties, other additives are added to these binders. The most widely used additives in recent years include boric acid, which was used in the previous studies by the authors [9,10], and phosphoric acid and its salts [11,12].

The addition of phosphorus-containing compounds has already been described in several publications [11,12,13,14]. Phosphoric acid is most often used in connection with so-called acid-based geopolymers, in which case phosphoric acid completely replaces the alkaline component of the material [7,14]. The combination of acidic and basic components to form alkali-activated aluminosilicates is less common. It is used, for example, by Hung et al. [15]. A strong alkaline solution allows only some addition of phosphoric acid before the phosphates precipitate. Therefore, a potassium phosphate solution was used in this article. According to available publications, which used nuclear magnetic resonance to study the composition of the material, phosphates are able to form bonds with aluminosilicates to form AlPO_4_, thereby altering the structure of aluminosilicates and their properties [16,17,18].

Furthermore, these materials can be filled with a variety of fillers. These fillers can be of different shapes and materials according to the required properties of the resulting material. In addition to reinforcing the binder, they can increase the heat resistance, heat capacity, plasticity, porosity or appearance of the material [2,19,20].

In recent years, much attention has been focused on the combination of inorganic materials and basalt or carbon fibers in various shapes, such as chopped fibers, roving or fabrics [20,21]. Fibers usually reinforce the organic matrix [22,23]. Conventional organic matrices cannot be used in high temperature applications due to their relatively low melting point (around 300 °C). This attribute of organic matrices limits the use of fibers, therefore the use of an inorganic matrix is appropriate. There have been some studies that examined inorganic composites precisely because of their resistance to high temperatures [3,20,21] and low emissions in case of fire [24,25]. Krystek et al. even compared mechanical properties of carbon/epoxy and carbon/geopolymer composites [26].

Inorganic materials based on alkali-activated aluminosilicates (called geopolymers under certain conditions) are used in many industries. The variability of their composition makes it possible to prepare materials with different densities and viscosities, depending on which the type of material application also differs. The material can be poured into a container/mold [20,27], sprayed [28], painted [10], extruded (3D print) [29,30] or used as a bath (pultrusion) [15]. The fabrics, used in this study, can be soaked or coated with a binder. Fabrics are immediately layered on top of each other after binder coating, or painted fabrics can be stored under given conditions and stacked on top of each other for some time [9].

This study is aimed at the investigation of changes in the mechanical properties of composites with an aluminosilicate matrix depending on the content of potassium phosphate as a source of phosphorus in the matrix. Five different amounts of potassium phosphate were added to the matrix and then used to prepare six-layer carbon composites in order to improve the mechanical properties of inorganic composites after the application of high temperatures and to expand their application in industry. Tensile strength and Young’s modulus were tested on composites cured at laboratory temperatures and after exposure to temperatures of 400 °C, 500 °C, 600 °C and 800 °C for 1 h. Weight loss and thickness changes of the composite samples were observed after exposure to 800 °C.

## 2. Materials and Methods

### 2.1. Materials

Inorganic matrix (A-matrix) based on alkali-activated aluminosilicates was prepared using commercial metakaolinite-rich material [31,32] produced by the calcination of kaolinitic claystone in a rotary kiln at cca 750 °C (České lupkové závody, a.s., Nové Strašecí, Czech Republic), silica fume (České lupkové závody, a.s., Nové Strašecí, Czech Republic), commercial potassium silicate with a molar ratio SiO_2_:M_2_O equal to 1.7 (Vodní sklo, a.s., Prague, Czech Republic), potassium hydroxide pellets (89.9%, Lach-Ner, s.r.o., Neratovice, Czech Republic), potassium phosphate anhydrous (Penta, s.r.o., Prague, Czech Republic) and distilled water. Above mentioned raw materials including carbon plain weave fabric with a mass per unit area of 200 g/m^2^ (Carbon fabric eSpread 200 CHT, Porcher Industries, La Voulte-sur-Rhône, France) were chosen as the composite reinforcement based on previous experience with alkaline matrices [9,10].

Chemical compositions of powdered raw materials were determined by X-ray fluorescence (XRF, Bruker S8 Tiger, Billerica, MA, USA) and their phase composition (Figure 1) was analyzed by X-ray diffraction system (XRD, Bruker D8 Advanced, Billerica, MA, USA). Potassium water glass was identified by an inductively coupled plasma optical emission spectrometer OPTIMA 8000 (PerkinElmer, Waltham, MA, USA) and the conventional acid-base titration method. The chemical compositions of used raw materials are shown in Table 1

### 2.2. Matrix

Six alkaline activators were prepared by mixing in each case the same amount of commercial potassium silicate, potassium hydroxide solution (37 wt%) and different amounts of powdered potassium phosphate. In order to facilitate dissolution, potassium phosphate was first mixed in a potassium hydroxide solution and then potassium silicate was added to the mixture. The amount of potassium phosphate was chosen so that its amount in the total matrix was 0 wt%, 2 wt%, 4 wt%, 6 wt%, 8 wt% and 10 wt%. Activators were stored in the fridge at 5 °C for 24 h. Then, the metakaolinite-rich material and silica fume were put in to the alkaline activators to form 6 matrices labelled after the potassium phosphate amount (M0P, M2P, M4P, M6P, M8P and M10P). The prepared mixture of matrix had a chemical composition of SiO_2_/Al_2_O_3_ = 33.9, K_2_O/Al_2_O_3_ = 3.98 and H_2_O/K_2_O = 12.1 (molar ratio). This matrix was mixed cca 30 min in a blender (Kenwood KVL8400S Chef XL Titanium, Havant, UK) and kept in a freezer at –18 °C for 24 h. This matrix composition was created and confirmed in published studies [9,10]. For simplicity, the preparation of matrices is schematically shown in Figure 2.

### 2.3. Laminate Composites

Six-layer composite plates were prepared by applying six types of matrices to carbon fabrics cut into 50 cm × 30 cm pieces. The matrices were applied to the fabric with a paint roller and the impregnated fabrics (prepregs) were individually placed between two pieces of plastic foil and stored in a freezer at –18 °C. After seven days, the prepregs were taken out of the freezer, stripped of plastic foil and stacked one by one (always in the same direction) to get the composite plate. Every prepared composite plate was placed between two pieces of peel-ply fabric, wrapped in plastic foil, compressed at 440 kPa for one hour and then cured in an oven at 65 °C for 3 h. After this time, the plates were unwrapped from the plastic foil and the peel-ply fabric and finally cured for 28 days under laboratory conditions. This method has been validated in previous studies [9,10]. The prepared composite plates were named analogously to the matrices using C0P, C2P, C4P, C6P, C8P and C10P. The percentage of filling of prepared plates is given in Table 2.

### 2.4. Pyrometric Cone Refractoriness

Pyrometric cone refractoriness of prepared matrices was determined by a heat microscope (Clasic CZ, type 0116 VAK, Řevnice, Czech Republic). European standard EN 993-12 was used as a basis for investigation of pyrometric cone refractoriness. This test was performed on a sharp-edged, trilateral cone with dimensions of 30 × 8.5 mm. Prepared test samples were studied and compared with the reference pyrometric cones. The test was carried out with a heating rate of 5 °C/min. The results were verified by three measurements.

### 2.5. Tensile Strength and Young’s Modulus

The prepared six-layer composite plates were cut into 250 × 25 mm samples by water jets (provided by AQUA-DEKOR s.r.o., Ústí nad Labem, Czech Republic). The ends of the samples were coated with epoxy resin (CHS-EPOXY 324 + P11, STACHEMA CZ s.r.o., Kolín, Czech Republic) and covered with sandpaper to protect the sample surface from sharp grips. Mechanical properties were determined by a universal testing machine LabTest 6.200 (LaborTech, s.r.o., Opava, Czech Republic). Tensile strength and Young’s modulus were measured according to the ASTM D3039 standard on the samples stored at room temperature (approximately at 25 °C) and on the samples that were exposed to temperatures of 400 °C, 500 °C, 600 °C and 800 °C for one hour in the oven (Clasic CZ, type 1018S, Řevnice, Czech Republic). The exposure temperature was added to the name of the composites (C0P-25, C0P-400, C0P-500, C0P-600, C0P-800, etc.).

## 3. Results and Discussion

### 3.1. Pyrometric Cone Refractoriness

The refractoriness of the tested matrices was examined by placing the tested cone between two nearest reference cones that melt simultaneously. From the results presented in Figure 3, it is evident that all examined matrices can withstand temperatures of up to 890 °C. Thus, they can be adequate substituent for high temperature applications and hence show their greatest advantage over the organic resins commonly used as matrices for composite materials [26].

Samples with a lower content of potassium phosphate in the matrix (0 wt%–4 wt%) began to melt at slightly higher temperatures than the other samples. This would mean that potassium phosphate does not improve the temperature resistance of the matrix. Although the samples with a higher amount of phosphate did not reach the highest temperatures, their volume remained almost unchanged, while the other samples swelled noticeably. Thus, the presence of potassium phosphate probably affects the chemical processes in the matrix, which dampen swelling due to the presence of silica fume.

### 3.2. Influence of Temperature on Composite Samples

The change in matrix volume was also evident with the composite samples, in which the samples with the highest amount of potassium phosphate were the least swollen. The high content of silicon dioxide in the form of silica fume was probably the main cause of the swelling of the prepared inorganic matrices. This phenomenon was described in the study by E. Prud’homme et al. [33]. Swelling of the matrix may be beneficial in certain circumstances. The change in the structure of the samples affects the mechanical properties of the material and its use. The advantage of a swollen matrix may be better thermal insulation properties; the disadvantage is that a higher brittleness of the matrix leads to worse mechanical properties. Those for which the swelling of the matrix was not significant will retain higher flexibility and tensile strength. Although the temperature affected the prepared types of composites differently, all samples withstood temperature of 800 °C. The increase in thickness of the composite samples after exposure to 800 °C for one hour are shown in Figure 4. The results reveal that an increasing amount of potassium phosphate in the matrix led to a reduction in swelling. Sample C0P, without the addition of phosphate, swelled by 32% compared to its original thickness. The addition of phosphate suppressed swelling by up to 27%. The swelling of the sample without an addition of phosphate (C0P) and the sample with the highest amount of potassium phosphate (C10P) are compared in Figure 5.

Because carbon burns at high temperatures, the matrix needs to tightly enclose the carbon fibers to protect them from oxygen access and combustion. The carbon fabric used in this study was placed in an oven at a temperature of 800 °C in air and after 1 h of exposure to this temperature the weight loss of the fabric was 80.7%. All prepared matrices and composite samples were dried at 110 °C, weighed and subjected to the same conditions as carbon fibers. The tested materials were reweighed after removal from the oven and their percentage weight loss was calculated. The results are shown in Figure 6.

Matrices with a lower amount of potassium phosphate and analogous composites made from these matrices had the highest weight loss. The dependence on the tensile strength test results was even seen in the course of the weight loss of the composite samples. It is clear that the more material left in the sample, the more its tensile strength is noted. However, we can also see in the picture a connection with the amount of potassium phosphate, which can affect the adhesion of the matrix to the fabric and is a better barrier to oxygen penetration.

### 3.3. Tensile Strength and Young’s Modulus

Figure 7 illustrates that the samples not exposed to higher temperatures showed almost comparable tensile strengths (324 MPa–373 MPa). Similar results can be seen at temperatures of 400 °C (277 MPa–302 MPa) and 500 °C (229 MPa–279 MPa), while the tensile strength decreases with increasing temperature. Thus, at normal temperature, the advantage of the presence of phosphors in the matrix was not observed.

Differences became evident when test samples were exposed to temperatures of 600 °C and 800 °C (28–87 MPa). While the results of pyrometric cone refractoriness show that samples with less phosphor content and the reference sample M0P resist higher temperatures better, the tensile strength showed the opposite trend at high temperatures. These samples showed several different parameters. The phosphate-free sample matrix appeared sintered and very brittle and broke easily during measurement. For these reasons, it was very difficult to measure composite samples. In contrast, samples with a higher amount of phosphate retained their flexibility, so the tensile strength of these samples was higher by up to 300%. Figure 8 shows characteristic fiber break with sample C10P after tensile strength. The matrix and carbon fiber together contributed to the tensile strength of the composite sample. The matrix of sample C0P was very brittle and was disrupted during the measurement.

The best tensile strength results are shown by composites with 6 wt% and 8 wt% potassium phosphate in the matrix (C6P, C8P). A lower amount of phosphate did not sufficiently affect the matrix to solidify the composite, and a higher amount of phosphate was probably already so high that the phosphate could not react with the aluminosilicate, this fact could lead to deterioration in mechanical properties. Katsiki et al. [11] tested geo-polymers with the addition of phosphoric acid. Even in this case, the compressive strength increased only to a certain limit value by the addition of phosphoric acid and then decreased again. The same was observed by the study by Zribi et al. [18]. The published results were measured at room temperature. However, it can be said that it is necessary to find a suitable amount of phosphorus compounds to improve the mechanical properties of alkali-activated aluminosilicates. The results of tensile strength even after temperature exposure are comparable with the previous publications. For example, Krystek et al. [26] stated that the tensile strength of their composites prepared from carbon fabric and inorganic matrix with the addition of boric acid was 150 MPa–190 MPa after exposure to a temperature of 400 °C and 127 MPa–132 MPa after exposure to 600 °C.

Figure 9 represents the effect of temperature and matrix composition on the stiffness of a laminate with an inorganic matrix. As in the case of tensile strength, we can also see that the content of potassium phosphate in the matrix does not have much effect on the Young’s modulus of the measured composites up to a temperature of 500 °C. Young’s modulus values remained above 34 GPa for samples exposed to only 25 °C. Exposure of the samples to temperatures of 400 °C and 500 °C caused a decrease in values between 20 GPa and 30 GPa. The more significant differences at higher temperatures (600 °C and 800 °C) were probably due to the embrittlement of the potassium phosphate-free matrix, as in the previous measurement. The values presented in Figure 7 and Figure 9 show that the embrittlement effect of the specimens was even more pronounced in the Young’s modulus than in the tensile strength measurement. Krystek et al. [26] also observed similar values of Young’s modulus at room temperature (33 GPa) and after exposure to temperatures of 400 °C (16 GPa) and 600 °C (19 GPa).

## 4. Conclusions

In this paper, the effect of potassium phosphate content in an aluminosilicate matrix on the mechanical properties of carbon prepreg composites was investigated. The following conclusions have been drawn from the obtained experimental results:Increased addition of potassium phosphate content in the matrix led to a slight decline in the refractoriness of inorganic matrix. The highest value of refractoriness (940 °C) was achieved for samples with 0 wt%–4 wt% of potassium phosphate in the matrix.Higher potassium phosphate amount resulted in a reduction in change of thickness of samples exposed to 800 °C for 1 h.A temperature of 800 °C led to a weight loss of all prepared matrices and composites. The addition of potassium phosphate reduced weight loss by 4.5% for the matrix and 10.6% for composites. Composites C6P and C8P showed the lowest weight loss.The tensile strength of the composite samples was up to 373 MPa. The tensile strength decreased after exposure to higher temperatures. It was proved that the addition of potassium phosphate suppressed this decrease. The C8P sample had 3× higher tensile strength than the sample without added potassium phosphate after 800 °C.Young’s modulus showed a similar trend. The addition of potassium phosphate suppressed a sharp drop in values after exposing the sample to high temperatures. The differences in Young’s modulus at a temperature of 800 °C were almost fourfold.

According to the results presented above, the addition of potassium phosphate significantly improved the properties of aluminosilicate composites. The increased amount of potassium phosphate led to the suppression of matrix swelling and weight loss of the samples. This fact was also related to the strengthening of composite samples, where the samples with a higher content of potassium phosphate showed higher tensile strength at high temperatures (600 °C and 800 °C). These results show that the prepared composites samples, in particular C6P and C8P, are suitable for high temperature applications. The prepared matrices are well suited for the preparation of so-called prepregs. The prepreg method makes it possible to divide the application of the matrix on the fabric and their subsequent layering to form a composite plate.

## Figures and Tables

**Figure 1 materials-15-00061-f001:**
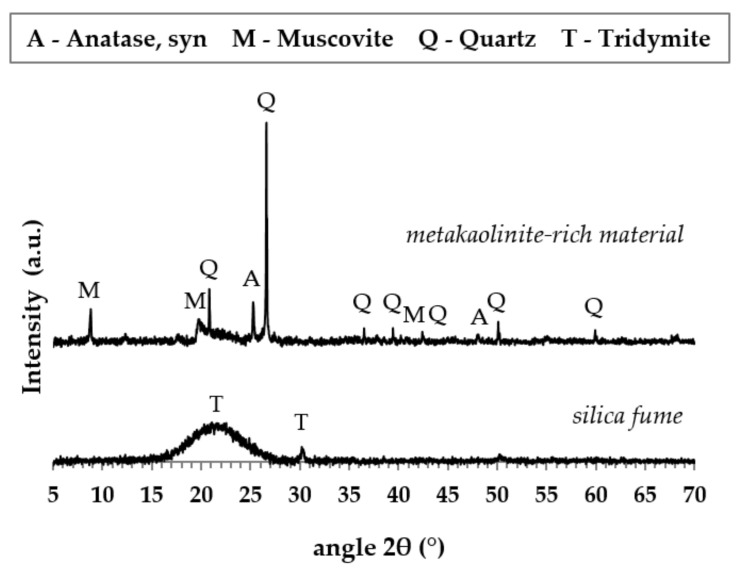
XRD patterns of metakaolinite-rich material and silica fume.

**Figure 2 materials-15-00061-f002:**
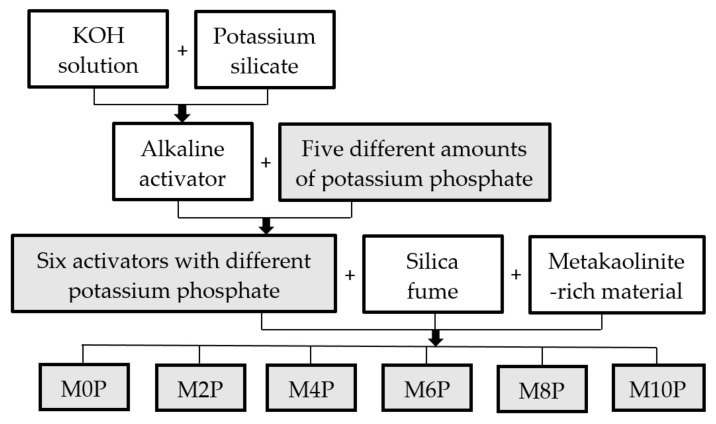
Scheme of six matrices preparation.

**Figure 3 materials-15-00061-f003:**
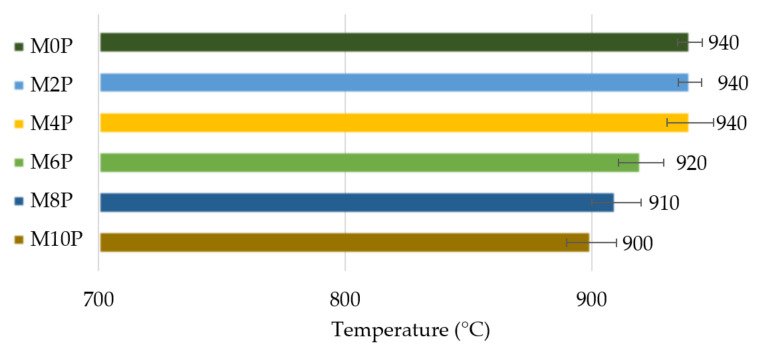
Pyrometric cone refractoriness of prepared matrices with different content of potassium phosphate.

**Figure 4 materials-15-00061-f004:**
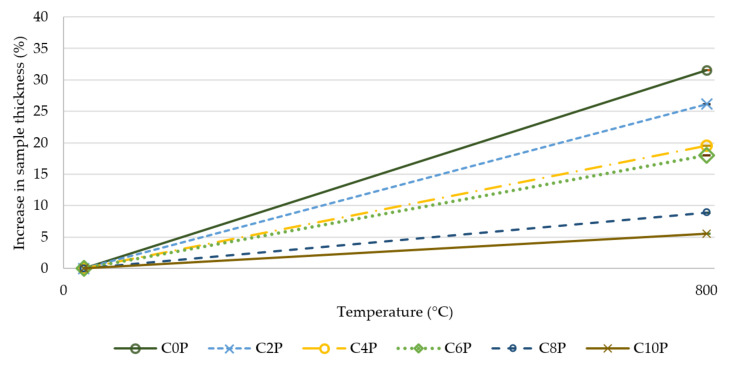
Increase in thickness (%) of composite samples after 800 °C.

**Figure 5 materials-15-00061-f005:**
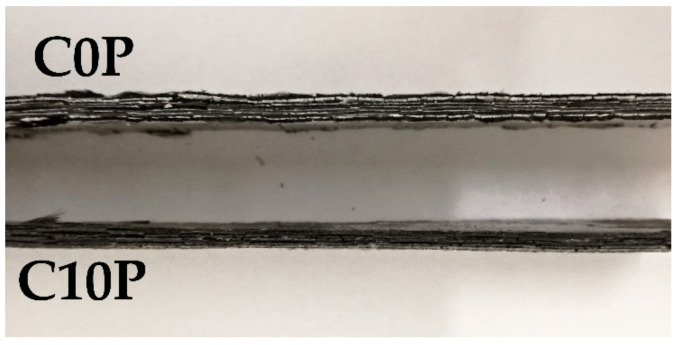
Difference in the thickness of samples C0P and C10P after 800 °C.

**Figure 6 materials-15-00061-f006:**
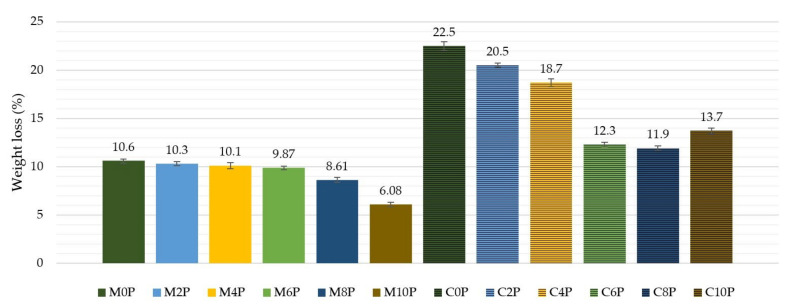
Weight loss (wt%) of matrices and composites after 800°C/1h exposure.

**Figure 7 materials-15-00061-f007:**
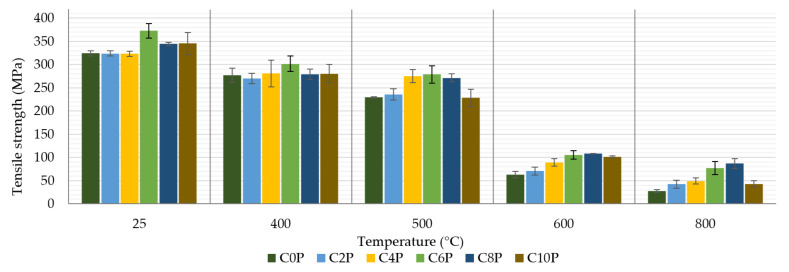
Tensile strength of prepared composite samples.

**Figure 8 materials-15-00061-f008:**
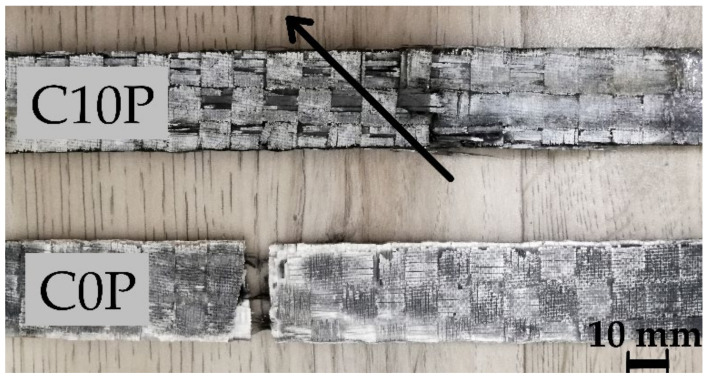
Characteristic fiber break with sample C10P after tensile strength and break of sample C0P with very brittle matrix.

**Figure 9 materials-15-00061-f009:**
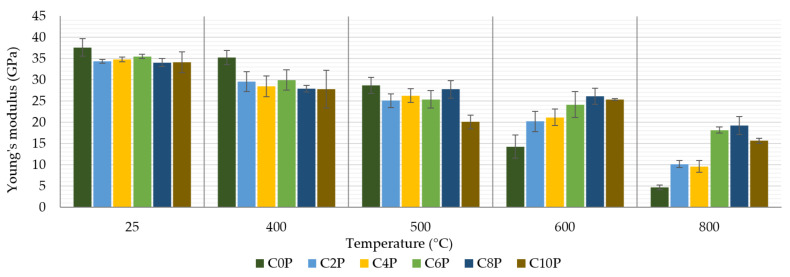
Young’s modulus of prepared composite samples.

**Table 1 materials-15-00061-t001:** Chemical composition of raw materials.

Material	Material Composition (%)								
	H_2_O	SiO_2_	Al_2_O_3_	Na_2_O	K_2_O	CaO	P_2_O_5_	Fe_2_O_3_	ZrO_2_
Metakaolinite-rich material	1.69	52.8	41.7		0.84	0.16	0.08	0.92	
Silica fume	0.99	96.4	0.42		0.04	0.09	0.42		1.27
Potassium silicate	63.9	18.9		0.23	16.7				

**Table 2 materials-15-00061-t002:** The percentage of filling of prepared composite plates.

Composite Plate	Area (m^2^) of 1 Piece of Carbon Fabric (200 g/m^2^)	Weight of Composite Plate (g)	Carbon Fabric Reinforcement (wt%)
C0P	0.15	436.9	41.2
C2P	0.15	443.3	40.6
C4P	0.15	451.1	39.9
C6P	0.15	439.0	41
C8P	0.15	429.6	41.9
C10P	0.15	445.5	40.4

## Data Availability

The data presented in this study are available on request from the corresponding author.

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
