# Peer review of "Effect of Potassium Phosphate Content in Aluminosilicate Matrix on Mechanical Properties of Carbon Prepreg Composites"

_materials, 2021, doi:10.3390/ma15010061_

Round 1

Reviewer 1 Report

Technical, grammatical and common mistakes are as follows

Comments for authors;

  • In the abstract part; the matrix was 0 %, no need of this %. This should be the standard one.
  • Write keywords in alphabetically order. Remove inorganic from it.
  • Introduction part. give rise to a more environmentally friendly (environment friendly).
  • Figure-1, manage it in Results and Discussion part.
  • Check the numbering of heading and sub-heading. Where is the heading #4?
  • %, °C, Figure, Table, Heading, Sub-heading, Numbers etc., write same format throughout the manuscript.
  • 3. Laminate Composites; write the name of curing agent. Also mentioned their purity, industry and country name.
  • 2. the author only claims that the volume increases of the composite samples after exposure to 800 °C ……. Is there any agglomeration appear in the samples at such high temp?
  • It will be better if provides some SEM images with Figure 4. to show more clearly the cracks on the matrix surface.
  • 2. All prepared matrices and composite samples were tested…… mentioned the sample weight?
  • Figure 7. Write the image magnification.

Warmly reminded; The present reference is not enough for this manuscript. The authors should want to add the latest and related references, besides the mentioned ones.

  • Cite the latest some of the mentioned references.
  • 234, no. 11-12, 2020, pp. 1759-1769. https://doi.org/10.1515/zpch-2019-1434
  • 2021, 13(2), 268; https://doi.org/10.3390/polym13020268
  • 233 (9):1233-1246. doi:10.1515/zpch-2018-1338 

Reviewer 2 Report

This paper is focused on the variation pattern of mechanical properties of aluminosilicate matrix composites with different potassium dihydrogen phosphate contents was investigated. The work is exhaustive and well supported by data. Guidance for improving the mechanical properties of aluminosilicate matrix composites. It is suggested that the research can be received after minor revise. The following are some suggestions for the authors. 1. In terms of narrative principles and processes, draw some flow charts to reduce unnecessary repetition in the middle paragraph. 2. The pictures in the article lack description or analysis, such as in Figure3 and Figure7. 3. The title of Figure 3 does not match the description of the Y axis. 4. Analyze and summarize the content of the conclusion to highlight the viewpoint.

Reviewer 3 Report

The work submitted for review is a scientific article. The work has 11 printed pages and has an appropriate editorial structure - divided into chapters and sub-chapters. The work is very interesting and important for the development of the scientific discipline of material engineering. It contains the results of research of utilitarian importance, important for industry. I have comments for work: Introduction: The authors broadly described the problems of inorganic additives to the matrix of the composite and their influence on the properties. The authors should develop issues related to the problems of high temperatures. They should describe why they intend to conduct the tests at the temperatures of the fire environment and on the basis of which work the test conditions were adopted. The last sentence in the Introduction chapter should be moved to Conclusions. I also propose formulating the purpose of the work. 

Materials and Methods:

Please add the percentage amount of potassium phosphate is given by weight or by volume (L: 111). 

Please provide the name and manufacturer of the water jet sampler (L: 144).

Results and discussion:

Please add reference to the information presented in L: 158-159. 

I propose the analysis of the research results to refer to the works of other researchers (L: 163-169). There is no in-depth scientific discussion of the research results. 

Figure 3 shows the changes in the thickness of the samples. The caption after the drawing is incorrect. The drawing information in L: 179 is incorrect. 

L: 185-190: The test method does not describe how the weight loss of the samples was measured. This information should be completed. 

I suggest expanding the scientific discussion. Searching in the databases of scientific journals and wider reference to the works of other scientists. In particular, I am asking for an analysis of the influence of the applied composites additives on their destruction mechanisms as well as the remaining strength and modulus of elasticity after thermal shock. 
